# Proportion of Respiratory Syncytial Virus, SARS-CoV-2, Influenza A/B, and Adenovirus Cases via Rapid Tests in the Community during Winter 2023—A Cross Sectional Study

**DOI:** 10.3390/diseases11030122

**Published:** 2023-09-15

**Authors:** Dimitrios Papagiannis, Garifallia Perlepe, Theodora Tendolouri, Polyxeni Karakitsiou, Georgia Damagka, Anna Kalaitzi, Sofia Alevra, Foteini Malli, Konstantinos I. Gourgoulianis

**Affiliations:** 1Public Health & Vaccines Laboratory, Department of Nursing, School of Health Sciences, University of Thessaly, 41110 Larissa, Greece; 2Respiratory Medicine Department, Faculty of Medicine, University of Thessaly, 41500 Larissa, Greece; perlepef19@gmail.com (G.P.); kgourg@uth.gr (K.I.G.); 3MD Private Sector of Greek Health System, Kanouta 1 Str. Trikala Thessaly, 42100 Trikala, Greece; tendolouridora@gmail.com; 4MD Private Sector of Greek Health System, Annas Komninis 1-3 Str. Trikala Thessaly, 42100 Trikala, Greece; p.karakitsiou@gmail.com; 5MD Private Sector of Greek Health System, Lefkosias Str. 3 Larissa Thessaly, 41335 Larissa, Greece; ntamagageo@yahoo.gr; 6Pediatric Department, General Hospital of Larissa, Tsakalof 1 Str. Larissa Thessaly, 41221 Larissa, Greece; akalait@hotmail.com (A.K.); alevrasofia59@gmail.com (S.A.); 7Respiratory Disorders Laboratory, Department of Nursing, University of Thessaly, 41110 Larissa, Greece; mallifoteini@yahoo.gr

**Keywords:** RSV, SARS-CoV-2, influenza A/B, adenovirus, rapid test, vaccinations

## Abstract

Respiratory infections constitute a major reason for infants and children seeking medical advice and visiting health facilities, thus remaining a significant public threat with high morbidity and mortality. The predominant viruses causing viral respiratory infections are influenza A and B viruses (Flu-A, Flu-B), respiratory syncytial virus (RSV), adenovirus and coronaviruses. We aimed to record the proportion of RSV, SARS-CoV-2, influenza A/B and adenovirus cases with rapid antigen tests and validate the results with RT-PCR assays of upper respiratory specimens with a wide range of viral loads and (co)-infection patterns in children. Clinical samples were collected from early symptomatic children (presenting with fever and/or cough and/or headache within 5–7 days). The surveillance program was conducted in five private pediatric dispensaries and one pediatric care unit, from 10 January 2023 to 30 March 2023 in central Greece. The total sample of specimens collected was 784 young children and infants, of which 383 (48.8%) were female and 401 were male (51.2%). The mean age of participants was 7.3 + 5.5 years. The sensitivity of the FLU A & B test was 91.15% (95% CI: 84.33–95.67%), and the specificity was 98.96% (95% CI: 97.86–99.58%). The sensitivity and specificity of the adenovirus and RSV test was {92.45% (95% CI: 81.79–97.91%), 99.32% (95% CI: 98.41–99.78%)} and {92.59% (95% CI: 75.71–99.09%), 99.47% (95% CI: 98.65–99.86%)} respectively. Lastly, the sensitivity of the SARS-CoV-2 test was 100.00% (95% CI: 79.41–100.00%) and the specificity was 99.74% (95% CI: 99.06–99.97%). We recorded a proportion of 14.3% and 3.44% for influenza A and B, respectively, followed by a proportion of 6.9% for adenovirus, a proportion of 3.7% for RSV, and finally, a proportion of 2.3% for SARS-CoV-2. The combination of a new multiple rapid test with multiple antigens will probably be a useful tool with a financial impact for health systems targeting the early detection and appropriate treatment of respiratory infections in emergency departments in primary health care facilities.

## 1. Introduction

Respiratory infections are a major reason for infants and children seeking medical advice and visiting health facilities, thus remaining a significant public threat with high morbidity and mortality. Upper respiratory infections are more common than those affecting the lower respiratory tract. Undoubtedly, there are limited options to reduce a viral outbreak in the absence of massive vaccination strategies, making the early detection of new cases crucial for the implementation of precautions that could halt spread to contacts and allow for the proper management of patients [1,2,3]. In this context, the development of new readily available rapid diagnostic tests (RDTs) with sufficiently high specificity and sensitivity for the prompt detection of contaminated persons in the community is a huge challenge for public health authorities [4].

According to the published literature, one of the major causes of childhood acute lower respiratory tract infection and a leading cause of hospital admissions among young children globally, is the respiratory syncytial virus (RSV) [5]. In more detail, three million RSV hospitalizations and thousands of deaths occur annually in children aged <5 years around the world. The vast majority of these hospitalizations and deaths are observed in low- and middle-income countries [6]. Recently, data from 28 European countries estimated that an average of 245,244 hospital admissions due to respiratory infections in children under five years were associated with RSV. Two out of three cases occurred among children aged less than one year [7].

The peak of adenovirus infections is recorded in winter and spring and, contrary to the flu, new cases occur throughout the year. They are responsible for 10% of all childhood fevers, and every child has at least one adenovirus infection by 10 years of age. The typical clinical presentation either resembles that of a common cold or includes croup, bronchitis, and pneumonia. In children, apart from respiratory infections, adenoviruses cause intestinal infections [8]. Respiratory infections, followed by intestinal ones, are reported as the most common means of adenovirus transmission. Usually, respiratory infections occur by way of close contact with infectious material from an infected person or surface. Secretions from the respiratory tract may contain the virus, and the adenovirus can survive for many hours on objects, such as doorknobs, lab surfaces, and toys. The second mode of transmission is intestinal and usually occurs by way of fecal–oral contact, associated with poor hygiene practices like washing hands, or from the ingestion of contaminated food or water [9,10,11,12].

Special populations, such as young children, are at increased risk of severe illness from influenza, or the flu. Influenza is a respiratory infection caused by influenza virus A, B, or C [13]. Viral infections of the lower respiratory system account for 100,000 cases and 100 associated deaths per million annually in children aged under five years. Influenza and pneumococcal pneumonia are responsible for the majority of deaths [14]. At the beginning of the 20th century, during the influenza pandemic, the highest attack rates of influenza were recorded in school-aged children of 5 to 18 years old [15]. On the contrary, the case series of the 2009 influenza A(H1N1) pandemic did not present data on children separately or reported small numbers of children with viral infections [16]. Children with chronic medical conditions or aged <5 years old are at high risk of serious complications and death from influenza. During the H1N1 pandemic, the Centers for Disease Control and Prevention (CDC) monitored child influenza deaths through its influenza-associated pediatric mortality reporting system. For the period April–August 2009, in the United States, 36 deaths were reported among children aged <18 years; 19% of them were aged <5 years, and 67% had one or more high-risk medical conditions [17]. Effective control of viral infection requires the elimination of infected cells to limit the production and spread of the virus, as well as to establish a specific immune memory directed against viral antigens [18]. The H1N1 pandemic influenza did not appear to cause more severe disease than seasonal influenza A. Asthma was a significant risk factor for severe disease among children with H1N1 pandemic influenza than among those with seasonal influenza [19].

The coronavirus disease 2019 (COVID-19) has impressively increased the burden on healthcare globally. In comparison with the worldwide pandemic, children and adolescents are characterized by mild clinical presentation and more propitious outcomes than adults [20]. Despite the fact that the vast majority of acute pediatric SARS-CoV-2 infections are generally estimated as mild, the associated postinfectious conditions, including pediatric inflammatory multisystem syndrome and ‘long COVID’ in children, are more complex and worth paying attention to [21].

Coinfections of respiratory viruses cannot be excluded as a hypothesis in differential diagnosis. Furthermore, the clinical signs and symptoms of viral infections are similar to those of bacterial infections. It is challenging for public health authorities to clinically distinguish bacterial and viral infections using a diagnostic method and identify coinfections from multiple viral pathogens. In conclusion, rapid test detection of viral pathogens could offer useful clinical information—and possibly reduce hospitalization cost [22].

The predominant viruses causing acute viral respiratory infections are influenza A and B viruses (Flu-A, Flu-B), RSV, and coronaviruses. A prompt and accurate diagnosis of viral infection can be challenging. Rapid and definite diagnosis of viral infections could improve clinical outcomes. Rapid antigen tests could offer quick and affordable results at the point of care, enabling a reliable detection of viral load samples. Therefore, early and proper diagnosis of respiratory infections is expected to reduce the inappropriate use of antibiotics and provide the possibility of using antiviral therapy [23].

The CDC, following recommendations from the agency’s independent vaccine advisory committee and approvals from the US Food and Drug Administration, has approved the use of two new RSV vaccines for older adults and expects them to be available in the next fall [24]. Vaccines are one of the most effective measures of public health, saving lives and pre-venting lifelong disabilities. The importance of vaccination and the introduction of new vaccines in the general population, and especially in vulnerable patients such as patients with autoimmune diseases, has been established in several studies [25].

In the present cross-sectional study, pending the new guidelines for the RSV vaccine and its probable introduction in the National Immunization Program targeting maternal and young children’s immunization, we aimed to record the proportion of RSV, SARS-CoV-2, influenza A/B, and adenovirus with rapid tests and validate the tests with RT-PCR assays. To this end, we used a convenient number of upper respiratory specimens with a wide range of viral loads and identified (co)infection patterns in young children visiting the private and public pediatric health sector in central Greece.

## 2. Materials and Methods

In this cross-sectional study, we used nasal specimens. The present surveillance program was conducted in five private pediatric dispensaries and one pediatric care unit department from 10 January to 30 March 2023. Clinical samples were collected from early-symptomatic children (presenting with fever and/or cough and/or headache within 5–7 days). The clinical samples were collected from the pediatric department of the General Hospital of Larissa, Greece, and five private pediatric dispensaries from early-symptomatic children. The parents or guardians of the participants were informed about the aims of the study by physicians. The age and sex of participants were reported as demographic details. The eligibility criteria to participate in the present study were the presence of symptoms of children (presenting with fever and/or cough and/or headache within 5–7 days) and parental consent. Participation in the study was optional, and parents signed a written consent after being fully informed about the aim of the study. Statistical analysis was performed using Excel 2019 (Microsoft, Redmond, Washington, DC, USA) and IBM SPSS (version 26) SPSS Chicago IL, and sensitivity and specificity with a 95% confidence interval (95% CI) were estimated based on binomial distribution.

A simplified specimen collection was introduced, since only one swab was required for the detection of four different pathogens. Our methodology, which involved the use of a single swab and consequently one extraction during a healthcare visit, as opposed to four distinct tests (samples), was designed to reduce discomfort, introduce a more cost-effective solution, reduce the demand on personnel, and eliminate unnecessary stress for all involved parties.

### 2.1. Principle of Assay

Antibodies specific to virus proteins are coated on the test line region of the nitrocellulose membrane. During testing, antigens of each virus in the specimen react with the antibodies that are coated onto gold nanoparticles. As the sample flows through the test membrane, it migrates up to react with the antibodies immobilized on the membrane and generate one colored line in the test region. The presence of this colored line indicates a positive result. To serve as a procedural control, a colored line will always appear in the control region if the test has been performed properly (Appendix A). The test results in this study were interpreted after 15 min.

After the sample selection, samples were stored at −20 °C and two aliquots were taken. Samples were anonymized and unique code numbers were given by the physicians to each pair of sample tubes. In this study, the clinical performance of four rapid antigen tests was compared to RT-PCRs of upper respiratory specimens from 784 underage individuals taken from January 2023 to March 2023. The four RDTs that were used in this study were manufactured by PROGNOSIS BIOTECH S.A (Larissa, Greece) and complied with the requirements of EN ISO 13485:2016. The tests that were used, and their respective functions, are presented below:Rapid Test FLU_COVID for the detection of influenza A/B and SARS-CoV-2 antigens in nasal or nasopharyngeal specimen (V16XX).Rapid Test FLU A_B for the detection of influenza A/B antigens in nasal or nasopharyngeal specimen (V17XX).Rapid Test RSV for the detection of respiratory syncytial virus antigen in nasal or nasopharyngeal specimen (V15XX).Rapid Test ADENOVIRUS for the detection of adenovirus antigens in nasal or nasopharyngeal specimen (V18XX).

### 2.2. Positive and Negative Predictive Values

We use diagnostic tests with the aim of classifying patients into two groups according to the presence or absence of disease. The main question is to quantify the ability of these binary tests to discriminate between patients who do or do not have the disease of interest. The results can be displayed in a 3 × 3. Table 1: the columns indicate gold standard results and the rows indicate rapid test results. The terms positive and negative allude to the presence or absence, respectively, of the condition of interest. The number of subjects with the condition testing positive and negative in rapid tests is indicated by *a* and *c*. The number of subjects without the condition testing positive and negative in rapid tests is indicated by *b* and *d*. The entire number of considered subjects ought to be *a* + *b* + *c* + *d*.

Diagnostic accuracy relates to the ability of a test to discriminate between the target condition and health. This discriminative potential can be quantified by measures of diagnostic accuracy such as sensitivity and specificity. Sensitivity and specificity are proportions; their confidence intervals can be computed utilizing the basic methods for proportions. Sensitivity is the proportion of true positives that are correctly identified by the test, given by:$$Sensitivity=\frac{True\;positives}{True\;positives+False\;negatives\;}=\frac{a}{a+c}$$

Specificity is the proportion of accurately distinguished subjects without the condition. It is the proportion of true negatives that are accurately recognized by the test:$$Specificity=\frac{True\;negatives}{False\;positives+True\;negatives\;}=\frac{d}{b+d}$$

Positive predictive value (PPV) and negative predictive value (NPV) are the other two fundamental measures of symptomatic precision. Positive predictive value (PPV+) is the proportion of patients with positive test results who are correctly diagnosed and is defined as:$$PPV=\frac{Sensitivity\;*\;\pi }{Sensitivity\;*\;\pi +\left(1-Specificity\right)\;*\;\left(1-\pi \right)}$$

Negative predictive value (NPV−) is the proportion of patients with negative test results who are correctly diagnosed:$$NPV=\frac{Specificity\;*\;\left(1-\pi \right)}{Sensitivity\;*\;\left(1-\pi \right)+\left(1-Sensitivity\right)\;*\;\pi }$$

Confidence intervals for sensitivity, specificity, PPV and NPV are also calculated [26].

### 2.3. Rapid Test Validation

In detail, two nasal swabs from 784 underage individuals were obtained simultaneously—the first from one nostril according to WHO guidelines for molecular analysis, and the second from the other nostril according to the manufacturer’s specifications for antigen testing [22]. Concerning molecular analysis, RNA/DNA extraction was performed using the NucleoSpin^®^ Virus Isolation Kit (MACHEREY-NAGEL Gmbh & Co. KG, Düren, Germany). For RT-PCR, the Real SARS-CoV-2/Flu/RSV (Operon S.A., Cuarte de Huerva, Spain) and ZENA HAdv qPCR Detection Kit (AMD Advanced Molecular Diagnostics, Nottingham, UK) were used to detect the above viruses, respectively, on a SaCycler 96 Real Time PCR system from Sacace Biotechnologies Srl.

## 3. Results

The total sample consisted of 784 young children and infants, of whom 383 (48.8%) were female and 401 were male (51.2%). The mean age of the study population was 7.3 + 5.5 years. In the present study, rapid test FLU A_B demonstrated high diagnostic accuracy regarding the detection of influenza A (Table 2). The sensitivity of the test was 91.15% (95% CI: 84.33–95.67%) and the specificity was 98.96% (95% CI: 97.86–99.58%). The positive predictive value (PPV) was 93.64% (95% CI: 87.54–96.86%), and the negative predictive value (NPV) was 98.52% (95% CI: 97.35–99.17%) (Table 3). Rapid test FLU A_B demonstrated high detection rates across most Ct ranges. For samples with Ct < 15, the detection rate was 100.00% (95% CI: 15.81–100.00%). For samples with 15 ≤ Ct < 25, the detection rate was 100.00% (94.64–100.00%). For samples with 25 ≤ Ct < 30, the detection rate was 100.00% (85.18–100.00%). For samples Ct ≥ 30, the detection rate was 52.38% (29.78–74.29%). The findings of this study suggest that rapid test FLU A_B has high sensitivity and specificity for the detection of influenza A. The high positive predictive value (PPV) and negative predictive value (NPV) indicate that the test has a high accuracy in both confirming the disease in patients who test positive and ruling out the disease in patients who test negative. These findings support the use of rapid test FLU A_B as a valuable diagnostic tool in clinical settings.

Secondly, we aimed to evaluate the diagnostic performance of rapid test FLU A_B for influenza B (FLUB) in a pediatric population (Table 2). The test demonstrated high sensitivity [91.67%, (95% CI: 73.00–98.97%)] and specificity [99.34%, (95% CI: 98.47–99.79%)]. The PPV was 81.48% (95% CI: 64.56–91.40%) and the NPV was 99.74% (95% CI: 99.01–99.93%) (Table 3). Rapid test FLU A_B demonstrated high detection rates across most Ct ranges regarding the detection of influenza B. For samples with 15 ≤ Ct < 25, the detection rate was 100.00% (75.29–100.00%). For samples with 25 ≤ Ct < 30, the detection rate was 100.00% (29.24–100.00%). For samples with Ct ≥ 30, the detection rate was 66.67% (9.43–98.30%). The findings of this study suggest that rapid test FLU A_B has high detection rates across most Ct ranges, indicating its utility in detecting influenza B in samples with varying viral loads.

In the present study, the rapid adenovirus test demonstrated high sensitivity and specificity in our study population (Table 4). The sensitivity of the test was 92.45% (95% CI: 81.79% to 97.91%), indicating a high ability to correctly identify patients with adenovirus. The specificity was 99.32% (95% CI: 98.41–99.78%), suggesting high accuracy in correctly identifying patients without the disease. The PPV was 90.74% (95% CI: 80.30–95.93%), indicating that among the patients who tested positive with the rapid adenovirus test, a high percentage truly had adenovirus. The NPV was 99.45% (95% CI: 98.61–99.79%), suggesting that among the patients who tested negative with the rapid test, a high percentage truly did not have the disease (Table 3).

The rapid adenovirus test demonstrated high sensitivity and specificity for the detection of adenovirus in our study population. These findings support its use as a valuable diagnostic tool. The rapid adenovirus test demonstrated high detection rates across most Ct ranges. For samples with Ct < 15, the detection rate was 100.00% (95% CI: 25.00–100.00%). For samples with 15 ≤ Ct < 25, the detection rate was 100.00% (95% CI: 90.97–100.00%). For samples with 25 ≤ Ct < 30, the detection rate was 88.88% (95% CI: 51.75% to 99.71%). For samples with Ct ≥ 30, the detection rate was 25% (95% CI: 6.31–80.58%). The findings of this study suggest that the rapid adenovirus test has high detection rates across most Ct ranges, indicating its utility in detecting adenovirus in samples with varying viral loads.

In accordance with the previous results, we also demonstrated high sensitivity and specificity for the rapid RSV test in our study population (Table 5). The sensitivity of the test was 92.59% (95% CI: 75.71–99.09%), indicating a high ability to correctly identify patients with RSV. The specificity was 99.47% (95% CI: 98.65% to 99.86%), suggesting high accuracy in correctly identifying patients without the disease. The PPV was 86.21% (95% CI: 70.04–94.35%), indicating that among the patients who tested positive with the rapid RSV test, a high percentage truly had RSV. The NPV was 99.74% (95% CI: 99.00% to 99.93%), suggesting that among patients who tested negative with the rapid test, a high percentage truly did not have the disease (Table 5). The rapid RSV test demonstrated high detection rates across most Ct ranges. For samples with Ct < 15, the detection rate was 100.00% (95% CI: 15.81–100.00%). For samples with 15 ≤ Ct < 25, the detection rate was 100.00% (95% CI: 80.49–100.00%). For samples with 25 ≤ Ct < 30, the detection rate was 83.33% (95% CI: 35.87–99.57%). For samples with Ct ≥ 30, the detection rate was 66.66% (95% CI: 9.43–99. 16%). The findings of this study suggest that the rapid RSV test has high detection rates across most Ct ranges, indicating its utility in detecting RSV in samples with varying viral loads. However, there was some variability in detection rates at higher Ct values, suggesting that the test may have reduced sensitivity in samples with lower viral loads.

The rapid test for SARS-CoV-2 demonstrated high sensitivity and specificity in our study population regarding the detection of SARS-CoV-2 (Table 6). The sensitivity of the test was 100.00% (95% CI: 79.41–100.00%), indicating a high ability to correctly identify patients with SARS-CoV-2. The specificity was 99.74% (95% CI: 99.06% to 99.97%), suggesting high accuracy in correctly identifying patients without the disease. The PPV was 88.89% (95% CI: 66.71–96.96%), indicating that among the patients who tested positive with the rapid SARS-CoV-2 test, a high percentage truly had SARS-CoV-2. The NPV was 100.00% (95% CI: 0.00% to 00.00%), suggesting that among the patients who tested negative with the rapid test, a high percentage truly did not have the disease (Table 6). The rapid SARS-CoV-2 test demonstrated high detection rates across most Ct ranges. For samples with 15 ≤ Ct < 25, the detection rate was 100.00% (95% CI: 54.07–100.00%). For samples with 25 ≤ Ct < 30, the detection rate was 100.00% (95% CI: 59.04–100.00%). For samples with Ct ≥ 30, the detection rate was 100.00% (95% CI: 29.24–100.00%). No samples with Ct < 15 were included in the study.

For the period of the present study, we recorded the highest proportion for influenza-A (110/784 × 100 = 14.3%), followed by the adenovirus, with a proportion of 54/784 × 100 = 6.9%, the respiratory syncytial virus, with a proportion of 29/784 × 100 = 3.7%, and finally influenza-B with a proportion of 27/784 × 100 = 3.44% and SARS-CoV-2 with an incidence proportion of 18/784 × 100 = 2.3%. We recorded high PPVs and NPVs for the diagnostic tests, which indicates that the available tests had high accuracy in both confirming the disease in the patients who tested positive and ruling out the disease in the patients who tested negative.

## 4. Discussion

In the present study, we showed promising results. On the one hand, we recorded the proportion of influenza-A and B, adenovirus, RSV, and SARS-CoV-2 infections in young children for a three-month period during the winter of 2023. On the other hand, we tested the specificity and sensitivity of rapid tests for these four respiratory diseases. This could be used in public health strategies as an implementation tool in order to control infections in the community.

The early diagnosis and isolation of infectious diseases in symptomatic patients to prevent the dissemination of the infection is very important, especially in emergency departments of hospitals and private health facilities [27]. The recent pandemic set a mandate for researchers and the scientific community for rapid, accurate and affordable SARS-CoV-2 diagnostic tools as a global priority. Diagnostic tests for infectious diseases, especially for those to which active acquired immunity via vaccination does not exist, are essential for widespread testing and contact tracing in order for public health authorities to control the spread of disease [28,29,30]. The gold standard method, which is the real-time reverse transcription polymerase chain reaction (RT-PCR), is a very sensitive method; however, it requires time. Rapid diagnosis tests, based on antigen detection, are faster, easier to perform and cost-effective [31].

Furthermore, rapid tests are best performed within the early stages of acute infection, when the viral load is at its highest levels (usually the first 5–7 days from symptom onset), after which antigen levels may drop significantly [32]. In the present study, we demonstrate that rapid tests offer the advantage of early detection of viral infections and help health professionals to start treatment in time as well as to reduce complications.

Given the experience of the recent global health emergency, policies and decisions from public health authorities targeted at limiting regular viral load activity in semi-closed communities, like workplaces, schools or universities, are based on general measures, such as basic hygiene measures, mask wearing, washing hands, and social distancing [33,34,35,36]. Moreover, proactive broad population surveillance to stop asymptomatic spread and prevent outbreaks is also advisable and has been implemented in many workplaces where the necessary budget was available [37]. According to WHO’s chief’s report at the 76th World Health Assembly “The end of COVID-19 as a global health emergency is not the end of COVID-19 as a global health threat. The threat of another variant emerging that causes new surges of disease and death remains, and the threat of another pathogen emerging with even deadlier potential remains [38]”. The diagnostic tests developed for SARS-CoV-2 constitute a critical component and a stable base to the overall prevention and control strategy for new threats and emergencies.

A plethora of published studies address the financial benefits of the early detection of infectious diseases. Towards this direction, a study by Paltiel et al. created a model which predicted that, without testing interventions, more than ten million infections, one hundred twenty deaths, and ten billion dollars in costs ($6.5 billion in hospital care and 3 $0.5 billion in lost productivity) would have occurred over a 60-day period in the SARS-CoV-2 pandemic in 2021 [34]. The results of this study support this view and emphasize the financial benefits of tests and early diagnosis of four viral respiratory infections in young children in winter 2023 [39].

Another important finding of the present study was the proportion of positive RSV samples. There are sparse data for the prevalence and incidence of RSV in the community in Greece. We present that 3.7% of the total sample were RSV positive. Results of older studies indicate that 61% of infants with bronchiolitis have had a documented RSV infection, with a case fatality rate that exceeds 0.7%, and that RSV is the most prevalent virus (56.6%) among children with a detected viral infection [40,41]. Similar results to the previous study were reported by Tsergouli et al. in a hospital in Northern Greece. More precisely, children under the age of 2 years, hospitalized for bronchiolitis, were tested for RSV infection during two RSV seasons (2016–2017 and 2017–2018). RSV was detected in 52.1% of patients, most of them younger than 6 months [42]. Similar data are reported in the results of a more recent study on RSV in young children by Tsagarakis et al. [43] Contrarily, a study designed for an adult population showed a low-level circulation of RSV during the autumn–winter period in 2021 [44].

Another retrospective study on RSV during a 12-year period (2002–2013) recorded a total prevalence of 27% of children testing positive for RSV infection [45]. In a serum epidemiological study of hospitalized children with atypical community-acquired pneumonia (CAP), IgM antibodies against RSV were detected in 20.7% of the total sample and coinfection was detected in three cases: two cases of mycoplasma pneumoniae and one of adenovirus [46].

The new guidelines dealing with RSV vaccination, after FDA authorization, state which age groups should be considered a priority for the vaccine. At the moment, the suggestions refer to the elderly. On the other hand, epidemiological data emphasize that in children less than one year, old RSV is the leading cause of hospitalization, and in children less than 5 years of age it is one of the principal causes of clinic appointments. Furthermore, it is estimated that RSV causes one million lower respiratory tract infections each year, resulting in a huge number of hospitalizations and being the most common cause of hospitalization in children under 5 years old [47,48].

In 2014, the WHO published a general guidance document that can be used as a reference for making decisions about the introduction of a vaccine into a national immunization program [49]. In line with this report, in the present study we present data about the incidence of RSV infections in young children. We also provide data that will be useful in the future to public health authorities in order to consider the introduction of the new RSV vaccine into special population groups expected to benefit from its implementation.

Additionally, the proportion of adenovirus infections in the present study was 6.9%. A previous study to determine the distribution of several respiratory viruses in children diagnosed with influenza-like illness during the winter periods in 2005–2008 also recorded similar results and the total adenovirus prevalence in that study was 7.5% [50].

Finally, according to the WHO, the 2022–2023 influenza epidemic season started prematurely in the European region. At the same time, concerns over RSV were rising and COVID-19 remained a threat. Along with COVID-19, these viruses were expected to have a high impact on health services and populations [51]. We recorded a low proportion of SARS-CoV-2 in a population of young children after three pandemic years, contrary to a modelling estimation of respiratory infections in the community in Wales for winter 2022–2023 [52]. A similar study in terms of methodology by Curatola et al. of children between 0 and 18 years old in a pediatric emergency department of a tertiary Italian hospital in the autumn and winter period of 2021–2022 demonstrated a total prevalence of SARS-CoV-2 of 12% [53].

The present study has several limitations. The main limitation is the study period. Data are lacking as the records system initiated after December 2022, making it impossible to determine the total proportion of infections for the winter period 2022–2023. Secondly, the sample of participants referred to the hospital and dispensaries may not be representative of the population, and findings may not apply to other groups. We used a convenience sample, meaning that the results cannot be generalized to the entire population. Furthermore, the absence of any technique aiming to sequence the influenza virus genome is included in the limitations of this study. Finally, we cannot exclude the possibility that mistakes in using the tests without supervision may have arisen, and that could have led to false results.

## 5. Conclusions

We present the proportion of four types of respiratory infectious diseases in a child population observed in the winter 2022–2023 period. Implementation of RDTs can improve the efficiency of early diagnosis of serious acute respiratory diseases, because RDTs are widely available and easy to use. There are no clinical signs or symptoms to distinguish circulating respiratory pathogens and most symptoms are common. Therefore, accurate laboratory diagnosis of respiratory secretions is necessary and is associated with a number of potential benefits. Early and confirmed diagnosis can prevent the need for empiric antibiotic therapy or allow treatment to be discontinued if already initiated, with many benefits for patients and healthcare systems. We tried to evaluate four RDTs for respiratory diseases. All the antigens presented high sensitivity and specificity for their four respective pathogens. The combination of a new multiple rapid test with different antigens will probably be a useful tool with a financial impact on early detection and appropriate treatment in emergency departments and in primary healthcare facilities. Finally, we provide data that could be useful for public health authorities to design measures against future threats of infectious diseases.

## Figures and Tables

**Table 1 diseases-11-00122-t001:** Relation between diagnostic test and presence or absence of disease.

Rapid Test	Real Time—Gold Standard Method		
	Positive (+)	Negative (−)	Total
Positive (+)	*a*	*b*	*a* + *b*
Negative (−)	*c*	*d*	*c* + *d*
Total	*a* + *c*	*b* + *d*	*a* + *b* + *c* + *d*

**Table 2 diseases-11-00122-t002:** Rapid test FLU A_B—influenza A and B.

Rapid Test Flu A_B	Real-Time PCR Influenza A		
	Positive	Negative	Total
Positive	103	7	110
Negative	10	664	674
Total	113	671	784
**Rapid Test Flu A_B**	**Real-Time PCR** **Influenza B**		
	**Positive**	**Negative**	**Total**
Positive	22	5	27
Negative	2	755	757
Total	24	760	784

**Table 3 diseases-11-00122-t003:** Rapid test FLU A_B—sensitivity and specificity.

Rapid Test FLU A	Mean Value	95% Confidence Interval
Sensitivity	91.15%	84.33–95.67%
Specificity	98.96%	97.86–99.58%
PPV	93.64%	87.54–96.86%
NPV	98.52%	97.35–99.17%
**Rapid Test FLU B**	**Mean Value**	**95% Confidence Interval**
Sensitivity	91.67%	73.00–98.97%
Specificity	99.34%	98.47–99.79%
PPV	81.48%	64.56–91.40%
NPV	99.74%	99.01–99.93%

**Table 4 diseases-11-00122-t004:** Rapid adenovirus test—sensitivity and specificity.

Rapid ADE Test	Real-Time PCR		
	Positive	Negative	Total
Positive	49	5	54
Negative	4	726	730
Total	53	731	784
**Rapid ADE Test**	**Mean Value**	**95% Confidence Interval**	
Sensitivity	92.45%	81.79–97.91%	
Specificity	99.32%	98.41–99.78%	
PPV	90.74%	80.30–95.93%	
NPV	99.45%	98.61–99.79%	

**Table 5 diseases-11-00122-t005:** Rapid RSV test—sensitivity and specificity.

Rapid RSV Test	Real-Time PCR		
	Positive	Negative	Total
Positive	25	4	29
Negative	2	753	755
Total	27	757	784
**Rapid RSV Test**	**Mean Value**	**95% Confidence Interval**	
Sensitivity	92.59%	75.71–99.09%	
Specificity	99.47%	98.65–99.86%	
PPV	86.21%	70.04–94.35%	
NPV	99.74%	99.00–99.93%	

**Table 6 diseases-11-00122-t006:** Rapid SARS-CoV-2 test—sensitivity and specificity.

Rapid SARS-CoV-2 Test	Real-Time PCR		
	Positive	Negative	Total
Positive	16	2	18
Negative	0	766	766
Total	16	768	784
**Rapid SARS-CoV-2** **Test**	**Mean Value**	**95% Confidence Interval**	
Sensitivity	100.00%	79.41–100.00%	
Specificity	99.74%	99.06–99.97%	
PPV	88.89%	66.71–96.96%	
NPV	98.52%	97.35–99.17%	

## Data Availability

The data that support the findings of this study are available on request from the corresponding author.

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
