# Peer review of "Proportion of Respiratory Syncytial Virus, SARS-CoV-2, Influenza A/B, and Adenovirus Cases via Rapid Tests in the Community during Winter 2023—A Cross Sectional Study"

_diseases, 2023, doi:10.3390/diseases11030122_

Round 1

Reviewer 1 Report

This manuscript by Papagiannis and colleagues presents a cross-sectional study that aims to record the incidence of RSV, SARS-CoV-2, Influenza, and Adenovirus using rapid tests and validate these tests with RT-PCR as the reference standard.

Despite its merits, this manuscript has several issues that require resolution prior to publication:

Major:

- The authors consistently use the term "incidence" throughout the manuscript, but they are actually reporting the proportion of positive patients among those tested, rather than calculating the true incidence rate, which involves new cases relative to the population and time. It is recommended to correct this misusage in the title, abstract, and manuscript itself.

- The methods section lacks crucial details, including the specific statistical analyses conducted. To enhance clarity, the authors should elaborate on how sensitivity, specificity, positive predictive value (PPV), and negative predictive value (NPV) were calculated. Additionally, the content found in lines 148-150 should be relocated from the methods section to the results section. Moreover, the manuscript should provide information about the process of patient recruitment, detailing whether patients were consecutive and if there were specific recruitment dates (e.g., between 10/1 and 30/3).

Minor:

- It is suggested to consider moving the detailed explanation of the rapid test procedure to a dedicated supplementary file. However, the final decision on this matter is left to the authors' discretion.

- Commendations are due for adhering to the STROBE Checklist. To enhance transparency, it is recommended to include a supplementary file that contains the completed STROBE Checklist, indicating where each checklist item was addressed in the manuscript.

- In the results section, the authors emphasize the significance of rapid and accurate diagnosis for influenza B in terms of patient management and control. This assertion appears somewhat out of context, implying that the diagnosis of influenza B is more pivotal than that of other viruses. If this is the case, the authors should provide further explanation and supporting references to substantiate this claim.

- During the PCR analysis for influenza, it would be interesting to report if you determined the lineages of influenza B, especially whether Influenza B/Yamagata was identified, given its apparent extinction since April 2021.

- In the introduction (line 71), it is advisable to incorporate recent and pertinent references concerning RSV-associated hospitalization in Europe (1).

Similarly, for adenovirus (lines 83-84), it is suggested to include recent and relevant references (2) to enhance the context.

- In lines 109-112 of the introduction, the notion that surveillance systems in many countries have been affected by the disruptions caused by COVID-19 is intriguing. To support this argument, the authors should consider citing an appropriate reference that underscores this point.

Please address these major and minor issues in the manuscript to ensure its readiness for publication.

1. https://pubmed.ncbi.nlm.nih.gov/37246724/

2. https://pubmed.ncbi.nlm.nih.gov/33230056/

Minor editing of English language required

Author Response

We would like to thank the reviewer for the thoughtful review of our manuscript. Your commitment to ensuring the accuracy and rigor of our study is deeply appreciated, and we are grateful for the opportunity to make these necessary revisions. Thank you once again for your time and dedication to advancing the quality of scientific discourse.

Reviewer 2 Report

The article is aimed at record the incidence of RSV, SARS-CoV-2, Influenza A/B and Adenovirus with Rapid tests and validate the tests with the RT-PCR assay in a representative number of upper respiratory specimens with a wide range of viral loads and identify (co)-infection patterns in young children visiting the private and public pediatric health sector in central Greece.

The authors wrote that pending the new guidelines for the RSV vaccine and its probable introduction in the National Immunization program targeting maternal immunization and young children.

In addition, for the purpose of the study, the authors mentioned: "in a representative number of upper respiratory specimens with ....." - unfortunately, the methodology does not contain any convincing description of the selection of the group - on what basis were these facilities selected? And there is absolutely no evidence that the group could be representative in any way;

Insufficient methodological description and lack of reference to the assumptions of the journal make me recommend the rejection of the article.

The results of the study, the incidence of individual viral diseases can be referred to the population of vaccinated / unvaccinated people. This would undoubtedly be the value of the article.

The authors do not indicate either the strengths of their study, nor do they indicate the limitations of the study, and these undoubtedly exist.

Author Response

We would like to thank the reviewer for the thoughtful review of our manuscript.

Reviewer 3 Report

The paper is interesting and well written. Specific immune response against each virus is characterized by cellular elements and a cytokine response. I suggest to discuss the role of Th17 cells (see and add as reference paper concerning Th17 and chronic inflmmatory immune-responses, Intern Emerg Med 6, 487–495 (2011). https://doi.org/10.1007/s11739-011-0517-7), and the role of IL-31/IL-33 axis, vitamin D and microbioma on immune responses (see and add as reference paper concerning these topics).  Finally, how many pediatric patients presented autoimmune diseases among those enrolled? The authors have to briefly discuss the importance of vaccinations against Sars-Cov2 and flu in patients with autoimmune diseases (see and add as reference paper concerning vaccinations in patients with systemic sclerosis and SLE).

The paper is interesting and well written. Specific immune response against each virus is characterized by cellular elements and a cytokine response. I suggest to discuss the role of Th17 cells (see and add as reference paper concerning Th17 and chronic inflmmatory immune-responses, Intern Emerg Med 6, 487–495 (2011). https://doi.org/10.1007/s11739-011-0517-7), and the role of IL-31/IL-33 axis, vitamin D and microbioma on immune responses (see and add as reference paper concerning these topics).  Finally, how many pediatric patients presented autoimmune diseases among those enrolled? The authors have to briefly discuss the importance of vaccinations against Sars-Cov2 and flu in patients with autoimmune diseases (see and add as reference paper concerning vaccinations in patients with systemic sclerosis and SLE).

Author Response

(The authors gave the same response as above.)

Reviewer 4 Report

I only have two suggestions for this manuscript:

1. Suggest title change, too long!

2. Line 187 need detailed description, what solution and how much solution was in the tube?

Author Response

(The authors gave the same response as above.)

Round 2

Reviewer 1 Report

Thank you for addressing (or answering to) all the comments. 
I think moderate editing of English language is required in the current version of the manuscript. 

Moderate editing of English language required

Author Response

Response: thanks, the reviewer for the comments. We tried to editing our manuscript. Professor Malli edited the final edition of the manuscript she is a native speaker of English.

Reviewer 2 Report

Thank you for your replies. Nevertheless, several issues still require clarification. The authors changed the word "representative" to "convenience" - what does it mean? how was the study group assessed? How do you know it's "convenience"?

The authors further replied that "no vaccine is currently available for active immunization against RSV and adenovirus for children." However, the manuscript is also applicable to influenza and Covid-19, and the results can be related to the immunization status of children against these diseases. What I mean is some tangible benefit from the manuscript, not just a description of the facts.

Author Response

Thank you for your replies. Nevertheless, several issues still require clarification. The authors changed the word "representative" to "convenience" - what does it mean? how was the study group assessed? How do you know it's "convenience"?

Response: We would like to thanks the reviewer for the comments. We fully understand his concerns. According to international literature Convenience sampling is often used when other types of sampling methods are hard or impossible to use because of time, cost, or other issues. Even though it can be a quick and easy way to get data it can also have biases and limitations that can affect how well the results can be used in the real world and how reliable they are. We agree that our sample does not include all the criteria of homogenous convenience sample and for this reason referred as a limitation by the authors to limitations section.

 The authors further replied that "no vaccine is currently available for active immunization against RSV and adenovirus for children." However, the manuscript is also applicable to influenza and Covid-19, and the results can be related to the immunization status of children against these diseases. What I mean is some tangible benefit from the manuscript, not just a description of the facts.

Response: We would like to thanks the reviewer for the comments. Our future plan of research for the upcoming winter is to include all the suggestions about the vaccinated and not vaccinated children in a randomized sample. 
